# Music-induced emotion flow modeling by ENMI Network

**Yunrui Shang**[1], **Qi Peng**[3], **Zixuan Wu**[3], **Yinhua Liu**[2]*

**1** School of Automation, Qingdao University, Qingdao, China, **2** Shandong Key Laboratory of Industrial Control Technology, Qingdao, China, **3** Institute for Future, Qingdao University, Qingdao, China

* liuyinhua@qdu.edu.cn

**Data Availability Statement:** All relevant data are within the manuscript and its Supporting information files.

**Funding:** Funders have no role in study design, data collection and analysis, publication decisions, or manuscript preparation. None of the authors

## Abstract

The relation between emotions and music is substantial because music as an art can evoke emotions. Music emotion recognition (MER) studies the emotions that music brings in the effort to map musical features to the affective dimensions. This study conceptualizes the mapping of music and emotion as a multivariate time series regression problem, with the aim of capturing the emotion flow in the Arousal-Valence emotional space. The Efficient Net-Music Informer (ENMI) Network was introduced to address this phenomenon. The ENMI was used to extract Mel-spectrogram features, complementing the time series data. Moreover, the Music Informer model was adopted to train on both time series music features and Mel-spectrogram features to predict emotional sequences. In our regression task, the model achieved a root mean square error (RMSE) of 0.0440 and 0.0352 in the arousal and valence dimensions, respectively, in the DEAM dataset. A comprehensive analysis of the effects of different hyperparameters tuning was conducted. Furthermore, different sequence lengths were predicted for the regression accuracy of the ENMI Network on three different datasets, namely the DEAM dataset, the Emomusic dataset, and the augmented Emomusic dataset. Additionally, a feature ablation on the Mel-spectrogram features and an analysis of the importance of the various musical features in the regression results were performed, establishing the effectiveness of the model presented herein.

## Introduction

As an art form, music serves as a conduit for expressing thoughts and emotions through the amalgamation and arrangement of sounds. It encompasses sonic components such as melody, harmony, rhythm, and timbre, collectively forming a musical composition through time structuring [1–3]. With the advancement of music technology, there is a growing interest in exploring the connection between music and emotions as a means of emotional expression. However, the impact of music on individuals' emotions is constrained by subjective and external factors. Each exhibits varying emotional granularity, and different people may elicit diverse emotional responses to the same piece of music. Therefore, it must be noted that building a general system for this aim is very challenging due to the uncertain delineation of human behaviors in diverse situations [4].

receive a salary from any of our funders. (Grant number:2020YFB1313600).

**Competing interests:** The authors have declared that no competing interests exist.

Music emotion recognition (MER) constitutes a process of using computers to extract and analyze music features, form the mapping relations between music features and emotional space, and recognize the emotion that music expresses [5]. It is important to note that MER primarily focuses on how music induces human emotions while minimizing the influence of irrational factors, making it a universal approach. Models for MER fall into two categories: traditional machine learning models and deep learning models. Previous studies have utilized traditional machine learning models such as Decision Trees [6], Hidden Markov Models (HMM) [7], and Support Vector Machines (SVM) [8] for speech emotion recognition. Despite significant progress, these models often struggle with emotion recognition accuracy. Yang et al. [9] were the first to employ regression methods in MER, directly predicting arousal and valence values for each music sample, thus circumventing the inherent ambiguity in traditional classification methods.

The development of deep learning has provided a new approach to MER research. Convolutional Neural Networks (CNNs) are significant deep neural networks broadly leveraged for emotional analysis and achieved terrific results [10]. Deep Convolutional Neural Networks (DCNNs) have outperformed SVMs in speech-emotion tasks [11–14]. Compared to CNNs, Recurrent Neural Networks (RNNs) exhibit stronger expressive abilities as they can process diverse sequence information across different time steps. Weninger et al. [15] captured time-varying emotions through music using RNNs. The $R^2$ statistic evaluated the performance, and results of 50% valence and 70% arousal were achieved [16]. Zhao et al. [17] utilized the melody direction of notes as a musical feature and employed RNNs to classify musical emotions, achieving high accuracy. Combining CNNs and RNNs can enhance the accuracy of MER tasks [18]. Devi et al. [19] developed a music genre classification system using transfer learning techniques, employing the spectrogram of a music frame as input, which RNNs and CNNs evaluated for treating and analyzing negative emotions. Yang et al. [20] utilized parallel Recurrent Convolutional Neural Networks (RCNNs) for music genre classification, underscoring the importance of time series models in such tasks.

With the development of deep learning models, researchers increasingly recognize the significance of temporal aspects in tasks. Long Short-Term Memory Networks (LSTMs) exhibit high sensitivity to time, effectively capturing the long-term dependencies in time series through cell states and gating mechanisms. Numerous studies have focused on continuous music emotion prediction after the emergence of the Emomusic dataset [21] and the DEAM dataset [22]. Deep Bidirectional Long Short-Term Memory (DBLSTM) Networks enhance temporal context processing and excel in regression tasks [23]. Li et al. [24] illustrated its efficacy by integrating a Bidirectional Long Short-Term Memory Network (BLSTM) with a fusion Extreme Learning Machine (ELM) for dynamic music emotion prediction. Pei et al. [25] introduced the Deep Bidirectional Long Short-Term Memory Recurrent Neural Network (DBLSTM-RNN) framework for unimodal and multimodal emotion recognition, contributing to the audiovisual multimodal emotion field. Dong et al. [26] proposed a novel Bidirectional Convolutional Recurrent Sparse Network (BCRSN) for continuous music emotion prediction. Notably, the Weighted Mixed Binary Representation (WHBR) method transforms the regression prediction process into a weighted combination of multiple dichotomous problems, significantly reducing training time and improving prediction accuracy.

Researchers have recently sought to incorporate attention mechanisms into the MER task to enhance accuracy further. Sanga et al. [27] markedly enhanced the predictive performance of arousal and valence by utilizing attention mechanisms in LSTM. Huang et al. [28] proposed the ADFF method for music emotion recognition. This method employs an end-to-end attention-based deep feature fusion mechanism, utilizing logarithmic Mel spectrograms as input to capture salient emotional features in music clips effectively. Zhong et al. [29] introduced the

CBSA model, which integrates CNN, BILSTM, and self-attention mechanisms. This model emphasizes the critical features of local music emotion, the serialization of information about music emotion from local features, and the global key points of music emotion. Improved regression results were observed in both the Emomusic and DEAM datasets. These studies highlight the importance of considering two conditions simultaneously to enhance the accuracy of dynamic music emotion recognition:

*a*) **A complete and adequate set of musical features.**

At present, music feature sets often exhibit feature redundancy and incorporating excessive features into the model may result in overfitting. This phenomenon causes the model to memorize the noise in the data during training, thereby limiting its generalizability.

*b*) **Models with perfect long sequence prediction capabilities.**

The current model still needs to be improved in handling long sequences. Additionally, it needs to consider multi-scale temporal feature learning, resulting in suboptimal performance and efficiency.

Limited studies in MER employ regression methods to investigate dynamic emotional prediction simultaneously. In this study, we referenced the concept of emotion flow introduced by the American psychologist Csikszentmihalyi [30] to depict the dynamic shifts in human emotions during music playback. Emotion flow represents a cognitive state enabling individuals to immerse themselves in specific emotions deeply. We also utilized a music hierarchical classification method to extract physical and music-theory-level features. We then employed an Efficient Net [31] to extract features from Mel spectrograms for feature fusion. This extraction approach significantly reduces feature dimensions, ensuring each dimension is representative and efficiently mitigating overfitting and computational complexity issues. Ultimately, upon identifying the limitations of the traditional Informer model within MER tasks, we endeavored to enhance it to better align with the demands of MER [32–34].

In summary, the main contributions of this paper are as follows:

- Emotions change under the influence of music in the dimension of time. So we model emotion flow recognition as a regression problem using time series.

- We extract music features from various perspectives. We utilize a hierarchical classification method to extract time series features in the time domain, considering both physical and music theory perspectives. Additionally, we extract Mel-spectrogram features in the frequency domain to address the limitation of poor generalization observed with time series features. This comprehensive approach significantly enhances regression accuracy and accelerates model convergence speed.

- We introduce an ENMI Network for predicting the continuous emotion of music. The Efficient Net is employed to extract Mel-spectrogram features, while Music Informer, an enhancement of Informer, is utilized. This enhancement involves the removal of Probsparse Self-attention and Self-attention Distilling, adding several Layer Normalization Layers, and replacing the complex decoder part with the LSTM Layer. The model is demonstrated to be lightweight and efficient.

## Music-induced emotion

### Emotion flow on Arousal-Valence emotional space

The Arousal-Valence model (A-V model) [35] is currently considered the most effective model in emotion computing. It differs from the emotional ring model proposed by Hevner

[36] and the discrete emotion model proposed by Ekman [37] in that it effectively quantifies emotion. It contains two dimensions: Arousal and valence. Arousal indicates the intensity of people's feelings. Valence indicates the degree of pleasure in the people's feelings [38]. Therefore, our study only discusses the variation in the magnitude of emotion values (Arousal and Valence) without considering the variation between emotion types.

As shown in Fig 1, Thayer et al. [39] made a more detailed division in the A-V model: HAHV(high arousal high valence), HALV(high arousal low valence), LAHV(low arousal high valence), LALV(low arousal low valence). Since the annotations in many music emotion datasets are based on such a division, we illustrate the flow process of the dynamic labeling annotations in the dataset in the A-V model. This process can be divided into two stages:

1) For a music feature vector at moment t: $x_t = [x_{t,1}, x_{t,2}, \ldots, x_{t,N}]$, where $x_t$ is the music feature vector at the $t$th time point, N is the number of features, and its 2-Norm can be denoted as follows:

$$\| x_t \|_2 = \sqrt{x_{t,1}^2, x_{t,2}^2, ..., x_{t,N}^2} \tag{1}$$

Let $\Delta y$ be the amount of change in the 2-Norm of the feature vector in the neighboring sampling time $\Delta t$:

$$\Delta y = \| x_{t+1} \|_2 - \| x_t \|_2 \tag{2}$$

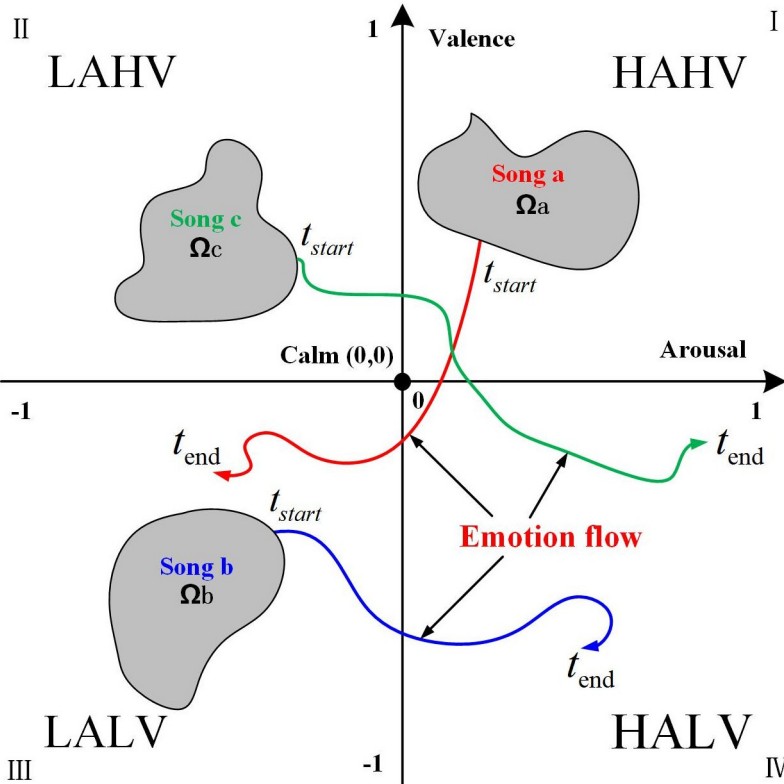

**Fig 1. Emotion flow on Arousal-Valence emotional space.**

and the accompanying change in the emotion values is denoted as:

$$\begin{cases} \Delta E_A(t) = E_A(t+1) - E_A(t) \\ \Delta E_V(t) = E_V(t+1) - E_V(t) \end{cases} \tag{3}$$

where $\Delta E_A(t)$ stands for the change of arousal dimension and $\Delta E_V(t)$ for the change of valence dimension. We use a region $\Omega$ to represent the unstable annotation segment of each music sample. With a Pearson Correlation Coefficient of $\rho_{(\Delta y, \Delta E)}$, and $\xi$, a very small positive constant that serves as a threshold for the generation of emotion flow, there is:

$$\Omega : |\rho_{(\Delta y, \Delta E)}| < \xi \tag{4}$$

The above process occurs in the early stage of music's effect on human emotions, and the emotions are not fully induced by music, which is unstable, so the correlation between the two is weak.

2) At this stage, the dominant role of music on emotion gradually increases, generating a music-induced emotional flow, and the emotional values are updated over time within the A-V model:

$$\begin{cases} A(t) = A_0 + f_A(t) + f_{AV}(V(t)) \\ V(t) = V_0 + g_V(t) + g_{VA}(A(t)) \end{cases} \tag{5}$$

where $A_0$ represents the initial arousal value, $A(t)$ represents the updated arousal value, $V_0$ represents the initial valence value, $V(t)$ represents the updated valence value, $f_A(t)$ and $g_V(t)$ represent the amount of emotion change in the music-induced arousal dimension and the valence dimension, respectively, which is the part of the deep learning model that needs to be learned. In addition, $f_{AV}(V(t))$ and $g_{VA}(A(t))$ represent the effects of the two emotions on each other, and this interaction is also important for human emotional change.

## Multivariate time series regression model

Multivariate time series regression aims to construct a regression model based on multiple time series. This problem involves a distinct dependency on numerous independent variables and observations within these time series. Specifically, when predicting the *ith* time series, it is essential to consider both the historical information of the *ith* time series and that of other time series. The objective of a multivariate time series regression model is to utilize multiple independent variables to elucidate and forecast specific time series.

Emotions require time to develop before they can be elicited. Similarly, once elicited, emotions typically endure for a particular duration [40]. Therefore, describing the MER problem as a multivariate time series regression problem is promising. As depicted in Fig 2, the emotional value (arousal or valence) within each predicted time window relies on various musical characteristics at the current moment, and the historical data of these characteristics and emotional values preceding the current moment. Overcoming these challenges requires an effective deep learning model capable of accurately forecasting changes in emotion flow.

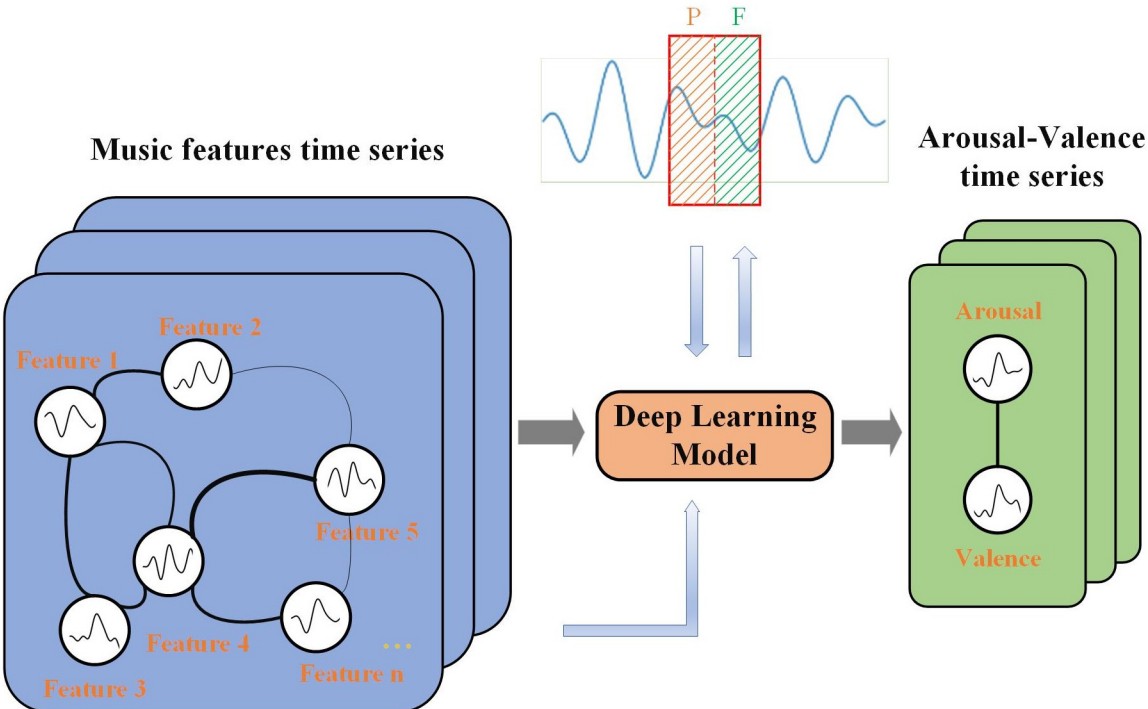

**Fig 2. Multivariate time series regression model (P:A historical window for emotion & F:A prediction window for emotion).**

## Model description

### Overview

In this section, our focus is on delineating the ENMI model's specific structure. As shown in Fig 3, this model comprises a feature extraction layer and a multivariate time series regression layer. To enhance the representativeness of extracted music features, we employed the Librosa Library to extract time series features at both the physical and musical levels of the music. Subsequently, an Efficient Net was utilized to extract music features from each music piece's Mel spectrogram. Once the Mel-spectrogram features were enhanced, they were jointly fed into the Music Informer Layer alongside the time series features to accomplish the regression task.

### Feature extraction

Zhou et al. [41] introduced a hierarchical classification method for categorizing musical characteristics. They argue that music exhibits physical acoustic attributes and semantic description distinctions resulting from the cognitive factors of music theory derived from artistic creation. Thus, they propose a classification framework comprising physical, musical, and cognitive levels. These levels correspond to music's low-level, medium-level, and high-level characteristics.

Table 1 illustrates the selection of features at various levels of music analysis. Spectral features, including MFCC, spectral centroid, spectral bandwidth, spectral flatness, spectral contrast, roughness, and time domain features like zero crossings, constitute the physical level of music, totaling 19 dimensions. Chromaticity characteristics, tempogram, and tonal centroid, totaling 14 dimensions, are characteristics of music theory. To address the limitations of

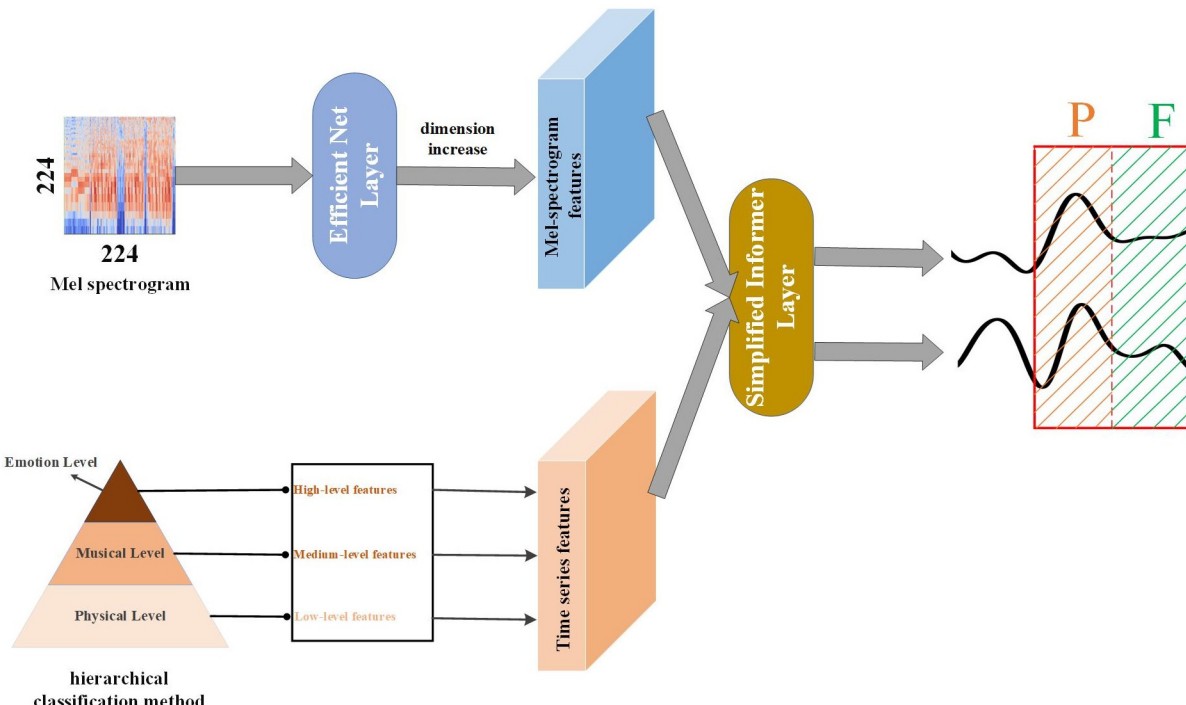

**Fig 3. The ENMI model ("dimension increase" adds a temporal dimension to the original tensor, which is used for fusion with time series features by transforming them into constant value series features).**

feature extraction using the Librosa Library and better integrate different feature types, deep learning techniques are employed to extract music features from Mel spectrograms, referred to as image features. Considering the complexity of cognitive features, emotional characteristics at higher levels are directly studied as observational measures.

**Table 1. Multi-level and multi-angle music feature extraction.**

| Level | Feature | Number of dimension | Method | Description |
|---|---|---|---|---|
| Physical Level | MFCC | 13 | Librosa Library | The degree to which the human ears perceive music at different frequencies. |
| | Spectral Centroid | 1 | | The frequency distribution of the sound signal and the center of gravity of the energy distribution. |
| | Spectral Flatness | 1 | | A measure of the uniformity or averaging of a spectrogram between different frequency components. |
| | Zero Crossings | 1 | | The number of times the sampled value of the sound signal passing through the zero point in each frame of the sound signal. |
| | Roughness | 1 | | Smoothness and uniformity of the sound. |
| | Spectral Contrast | 1 | | Information about an audio signal's relative strength and dynamic range of different frequency components. |
| | Spectral Bandwidth | 1 | | The frequency range in which energy is included in the signal spectrum. |
| Musical Level | Chromaticity Characteristics | 12 | | The energy in twelve scales over a period of time. |
| | Tempogram | 1 | | Rhythm and temporal structure in music. |
| | Tonal Centroid | 1 | | Tonal and harmonic structures in music. |
| Synthesis | Image Features | 10 | Efficient Net | Extracted by Mel spectrogram. |

## Efficient Net Layer

The Efficient Net model [31] is regarded as the leading CNN model to date, as it scales the depth, width, and resolution of Convolutional Neural Networks through web search technology, thereby significantly enhancing network performance. Notably, the Efficient NetB0 Network represents the authors' first lightweight model based on the baseline model, and we employed it to extract the Mel-spectrogram features.

Previous studies [5, 14, 26, 28, 42, 43] have demonstrated the effectiveness of spectrograms in MER tasks. A spectrogram illustrates the intensity or energy of various frequency components over time. Typically, the horizontal axis of a spectrogram denotes time, the vertical axis represents frequency, and color or brightness indicates the energy or intensity corresponding to each frequency. While spectrograms offer high resolution for frequencies, the perceived distance between different frequencies is not uniform. In contrast, the Mel spectrogram employs the Mel scale, nonlinearly transforming the frequency axis to better align with human auditory perception and accurately reflect differences in pitch perception. We utilized the Mel spectrogram as input for the Efficient Net Network. (Details of this process are described in the Experiment and Process sections).

The Efficient NetB0 Network comprises three parts: the first part features a standard Convolutional Layer with a 3×3 convolutional kernel size and a stride of 2, followed by a stack of 7 MBConv modules in the second part, and the third part includes a standard 1×1 Convolutional Layer, an Average Pooling Layer, and a Fully Connected Layer. Among the MBConv modules, the first is MBConv1, while the subsequent are MBConv6s. Additionally, the convolutional kernels used in all MBConv modules vary in size, with dimensions of 3×3, 3×3, 5×5, 3×3, 5×5, 5×5, and 3×3. Convolutional operations are used to extract local features [44]. The input model size for the Mel spectrogram is 224×224, and the output model yields a tensor of 1×10. By combining feature fusion and skip connection mechanisms, the network effectively extracts information at various levels, which proves advantageous for Mel-spectrogram feature extraction.

## Music Informer Layer

**Informer Layer.** The Informer model [32] is used to solve the problem of long-series time prediction. The model can effectively capture the accurate long-range dependent coupling between output and input, which makes up for the autoregressive limitations, quadratic time complexity, and high memory usage of the Transformer Encoder-decoder Architecture. As shown in Fig 4, in order to make the Informer more suitable for MER regression tasks and lightweight enough, we have also modified the Informer model (from now on referred to as Music Informer).

For long time series, with a rolling forecast setup with a fixed window size, the input at time point $t$ is:

$$X^t = \{x_1^t, x_2^t, \ldots, x_{L_x}^t | x_i^t \in R^{d_x}\} \tag{6}$$

where $L_x$ represents the length of the current input sequence, each point is a vector of the $d_x$ dimension, and the content of the vector includes all the music audio features and music emotion features in this window. The output corresponds to a prediction sequence that can be expressed as:

$$Y^t = \{y_1^t, y_2^t, \ldots, y_{L_y}^t | y_i^t \in R^{d_y}\} \tag{7}$$

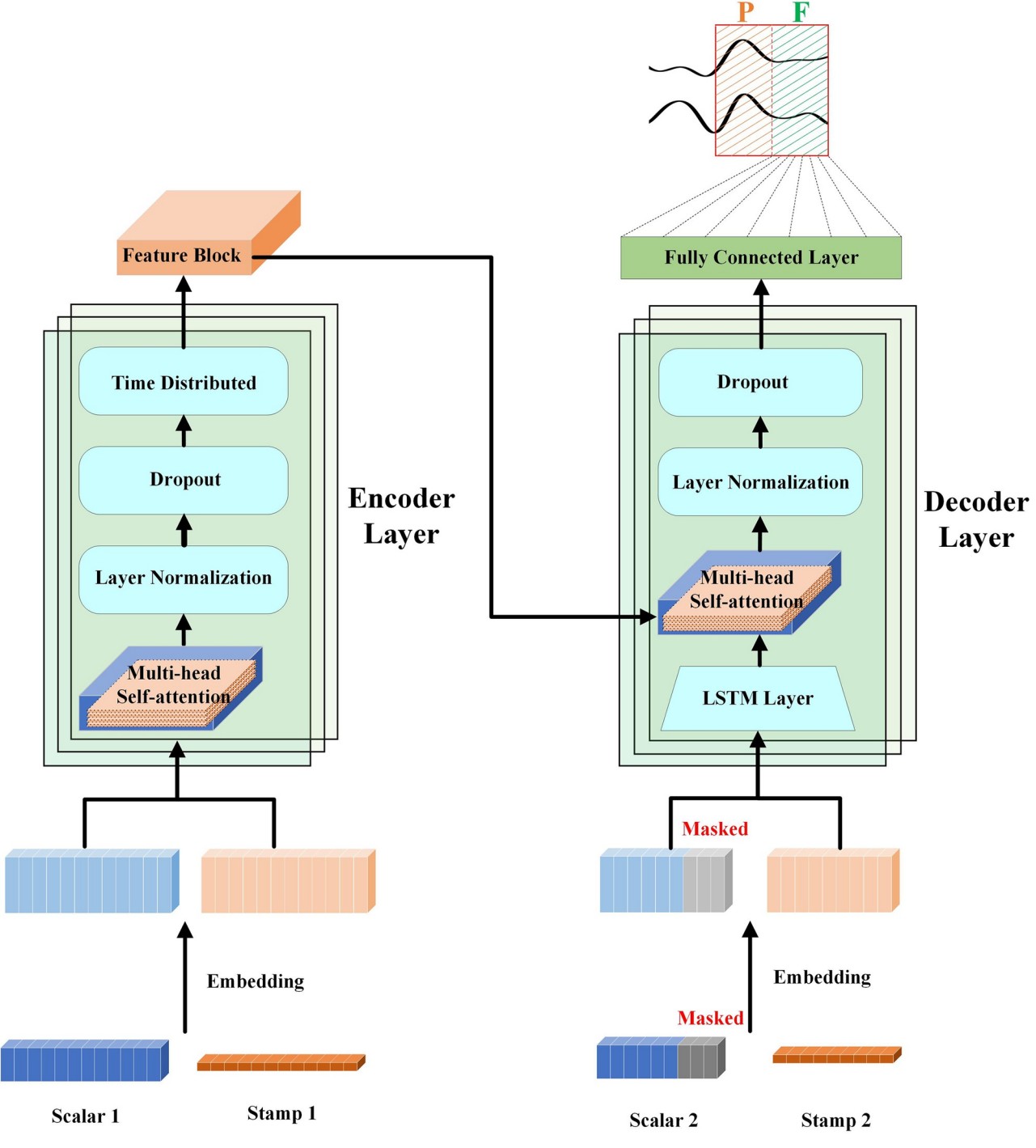

**Fig 4. Music Informer model.**

where $L_y$ is the length of the current output sequence, and each point is a vector of the $d_y$ dimension.

Before the time series data is sent to the Music Informer, the features of the time series and the Mel-spectrogram feature series extracted by Efficient Net need to be fused to expand the

Mel-spectrogram feature series into a constant value in the time dimension and merge the time series into a feature matrix as the input of the Music Informer model. First, the Embedding Layer is as follows:

$$\chi_{feed[i]}^{t} = \alpha u_i^t + PE_{(L_x(t-1)+i)} + \sum_p [SE_{(L_x(t-1)+i)}]_p \tag{8}$$

which consists of two parts, namely, Token Embedding and Positional Embedding. Token Embedding converts the sequence length into 512 dimensions that can enter the model through Conv1d. Positional Embedding is used to help the model understand the temporal order information of the input sequence. Typically, the sine cosine position is encoded:

$$PE_{(pos,2i)} = sin\left(pos/10000^{2i/d_{model}}\right) \tag{9}$$

$$PE_{(pos,2i+1)} = cos\left(pos/10000^{2i/d_{model}}\right) \tag{10}$$

where $pos$ is the location, $i$ is the index of the embedded dimension, and $d$ is the embedded dimension. Positional coding allows the model to encode information at different locations in a sequence to understand the chronological sequence better.

**Multi-head Attention Layer.** In the Multi-head Attention Layer, the query matrix, key matrix, and value matrix are calculated first. Then, by calculating the attention weight, the value matrix is weighted and summed to obtain the attention output, and the specific calculation formula is as follows:

$$Attention(Q, K, V) = softmax\left(\frac{QK^T}{\sqrt{d_k}}\right)V \tag{11}$$

where $d_k$ represents the dimension of the key in each attention head, this step calculates the similarity between the query and the key and weights the sums of the values accordingly.

For multiple attention heads (usually $h$), the attention mechanism is applied in parallel to different representation subspaces, and the output of each head is stitched together:

$$Multihead(X) = Concat(head_1, head_2, \ldots, head_h)W^O \tag{12}$$

$$head_i = Attention(XW_i^Q, XW_i^K, XW_i^V) \tag{13}$$

where $W_i^Q$, $W_i^K$, and $W_i^V$ are the weight matrix of the $ith$ attention head, and $W^o$ is the weight matrix used to stitch together the output of the multi-head attention.

While the Probsparse Self-attention in Informer significantly reduces computational requirements, it also results in the loss of crucial information. Given that the MER regression task relies on second-level prediction and involves a small number of features, it exhibits high sensitivity to feature localization and dependence on data. Therefore, we computed all attention parameters without employing the Probsparse Self-attention and Self-attention Distilling.

The Music Informer model comprises an encoder and a decoder. The encoder is tasked with feature extraction and encoding of input sequences. It includes multiple encoder layers, each incorporating a Multi-Head Attention layer, a Layer Normalization Layer, a Dropout Layer, and a Time Distributed Layer. On the other hand, the decoder is responsible for sequence prediction utilizing the features generated by the encoder. It comprises multiple decoder layers, each with an additional LSTM layer compared to the encoder layer. Furthermore, the feature extraction outputs from the encoder's Time Distributed Layer must be fused before inputting into the decoder's Multi-head Attention Layer.

**Time Distributed Layer.** The Time Distributed Layer is commonly employed for processing time series data and is typically utilized following the CNN Layer, RNN Layer, or LSTM Layer. This approach can effectively diminish the model's complexity and parameter count while enhancing its generalization ability. Its fundamental concept involves applying the same operation within the same layer to each time step in the sequence. This enables the model to learn time-dependent features across the entire sequence rather than solely focusing on individual time steps. For a sequence of data:

$$X = \{x_1, x_2, \ldots, x_i\} \tag{14}$$

where $x_i$ represents the eigenvector of the *ith* time step, and *i* denotes the length of the sequence. The Neural Network Layer employing time distributed will represent the input data $X$ as a matrix of $T{\times}D$, where $D$ represents the dimension of the eigenvector for each time step. Operations within this layer will subsequently be applied to the input eigenvector *i* at each time step $x_i$, producing the output $y_i$.

Specifically, let *f* denote the operation of the Neural Network Layer. The operation employing the Time Distributed Layer can be represented as:

$$y_i = f(x_i; \Theta) \tag{15}$$

where $y_i$ denotes the output corresponding to the *ith* time step in the input sequence, and $\Theta$ represents layer's parameters. It is crucial to emphasize that *f* denotes the identical operation applied at each time step, implying that the operation will be consistently applied to the input data at each time step, irrespective of the sequence length.

## Experiment and process

### Datasets

Media Eval Emotion in Music is a dynamic MER algorithm evaluation. The sample dataset applied in this evaluation is the result of the research and development of Mohammad Soleymani, among other researchers [45]. The Emomusic and DEAM datasets were employed to validate the reliability of the ENMI model for the multidimensional time series regression task. The Emomusic dataset comprises 744 music clips, each lasting 45 seconds. Every music segment has static A-V data labels and dynamic A-V time series labels annotated on the time domain. Moreover, the dynamic labels are annotated in each time bucket of 0.5 seconds from the 15th second until the end of the 45th second. The annotation process involved at least 10 individuals per snippet and was conducted by the Amazon Mechanical Turk crowdsourcing staff. The dataset DEAM, derived from the expansion of the Emomusic dataset, follows the same annotation style used in this study with 1802 music clips, each 45 seconds long. The tracks, audio, and emotional annotations for this dataset are publicly available.

58 longer music clips were removed from the DEAM to ensure uniformity in the size of the model dataset, hence limiting our focus to music clips elapsing 15s to 45s. (Access to the Emomusic dataset: https://cvml.unige.ch/databases/emoMusic/) (Access to the DEAM dataset: https://cvml.unige.ch/databases/DEAM/).

### Preprocessing

In order to comprehensively validate the generalization of the model, we incorporated 2×744 audio-enhanced samples into the Emomusic dataset to form a new dataset called the Augmented dataset. Moreover, all the enhancement data are drawn from the samples in the Emomusic dataset. The particular enhancement techniques used are shown in Fig 5. Both the

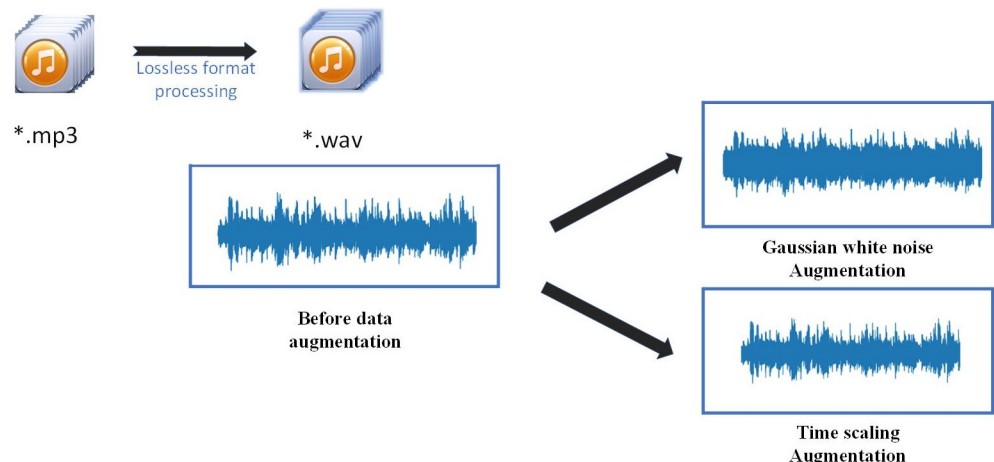

**Fig 5. Preprocessing and data augmentation.**

Augmented dataset and the DEAM dataset contain samples from the Emomusic dataset. Therefore, the effect of the difference in the number of samples between the two original datasets can be compensated through the integration of the Augmented dataset. This approach was similarly employed by Bhuwan Bhattarai in his study [46].

As illustrated in Fig 5, the regression task pertaining to time series prediction is highly sensitive to time scale. Moreover, the time acquisition window for dynamic data within the dataset is narrow, making it susceptible to generating corrupted data through over-processing. Consequently, after converting every MP3 file in the dataset into a lossless and uncompressed WAV file, noise enhancement and specific time-stretching were performed with augmenting data.

To address the risks of audio data label contamination by high-intensity white noise, the added intensity of white noise was set within a reasonable range. The intensity of the added white noise is related to the intensity of the original audio, as illustrated below:

$$std = (0.1 + 0.2 * e) * 1/20 \tag{16}$$

where $std$ represents the intensity of the added Gaussian white noise, and $e$ denotes the amplitude intensity of the music. Regarding time scaling enhancement, the original audio underwent time stretching within the range of [0.95, 1.05] magnification in the time domain. These two aforementioned processing methods did not alter the associated labels.

## Experimental setup

Following preprocessing, each audio file was segmented into non-overlapping subsegments of 500ms using the Librosa Library to align with arousal and valence annotations in the dataset. Each subsegment was converted into a Mel spectrogram using a sliding window with a frame size of 50ms and a step size of 10ms. The Mel spectrogram was then transformed into a three-channel image and normalized to [0, 255]. Subsequently, after conversion to a PIL image, its dimensions were adjusted to 224×224, and its pixel values were scaled to achieve a distribution with a mean of 0 and a standard deviation of 1. Serving as input to the Efficient Net model, this preprocessing facilitated more accessible pattern learning and enhanced training stability.

The training was conducted using the TensorFlow framework with the Adam optimizer, setting the learning rate to 0.001. The batch size was set to 128. Mean Squared Error (MSE)

was the loss function, while Root Mean Square Error (RMSE) was adopted as the primary metric. Recognizing the necessity of accountability for each sample in the test set and the susceptibility of RMSE to outliers and significant errors, Mean Absolute Error (MAE) was introduced as an additional metric. Furthermore, two callback functions, *ReduceLROnPlateau* and *EarlyStopping*, were employed. *ReduceLROnPlateau* function dynamically adjusted the learning rate by monitoring validation loss; if the loss didn't improve for 5 consecutive epochs, the learning rate was halved, with a minimum value of 1e-8. *EarlyStopping* ensured that training was terminated if the loss did not improve for 10 consecutive epochs, and the best weights were restored to prevent overfitting.

The training set was 90% of the entire dataset, with 20% for validation. The training and test sets were carefully balanced to include all song types, ensuring that the distribution difference between classes did not exceed 20%.

## Model architecture

Table 2 provides an overview of the architecture of Music Informer. In the Input Layer, the input dimensions are set as (*None*, 2, 45), where "*None*" indicates that the model can accept a variable number of samples, enabling it to handle input sequences of varying lengths.

The core architecture of the model comprises an encoder-decoder structure with two layers each for both the encoder and decoder and the head attention parameter is set to 8. Each sample comprises 2 time steps (adjustable window size) and 45 features. The model includes 4 Multi-head Attention Layers: *multi_head_attention*_4 receives inputs and performs the attention mechanism, *multi_head_attention*_5 takes outputs from the previous layer as inputs, *multi_head_attention*_6 receives outputs from the LSTM layer as inputs, and *multi_head_attention*_7 accepts the output of another LSTM Layer as input. Following each Multi-Head Attention Layer, there is a Layer Normalization Layer for output normalization.

**Table 2. Parameter model table (*When using "Window size = 2" to predict an emotional dimension*).**

| Type (layer) | Output Shape | Parameter | Connected to |
|---|---|---|---|
| Input Layer (input_2) | (None, 2, 45) | 0 | —— |
| Multi-head Attention (multi_head_attention_4) | (None, 2, 45) | 93741 | ['input_2 [0][0]', 'input_2 [0][0]'] |
| Layer Normalization (layer_normalization_4) | (None, 2, 45) | 90 | ['multi_head_attention_4 [0][0]'] |
| Dropout (dropout_4) | (None, 2, 45) | 0 | ['layer_normalization_4 [0][0]'] |
| Time Distributed (time_distributed_3) | (None, 2, 512) | 23552 | ['dropout_4 [0][0]'] |
| Multi-head Attention (multi_head_attention_5) | (None, 2, 512) | 1050624 | ['time_distributed_3 [0][0]', 'time_distributed_3 [0][0]'] |
| Layer Normalization (layer_normalization_5) | (None, 2, 512) | 1024 | ['multi_head_attention_5 [0][0]'] |
| Dropout (dropout_5) | (None, 2, 512) | 0 | ['layer_normalization_5 [0][0]'] |
| Time Distributed (time_distributed_4) | (None, 2, 512) | 262656 | ['dropout_5 [0][0]'] |
| LSTM (lstm_2) | (None, 2, 512) | 2099200 | ['time_distributed_4 [0][0]'] |
| Concatenate (concatenate_2) | (None, 2, 1024) | 0 | ['time_distributed_4 [0][0]', 'lstm_2[0][0]'] |
| Multi-head Attention (multi_head_attention_6) | (None, 2, 1024) | 2099712 | ['lstm_2 [0][0]', 'lstm_2 [0][0]'] |
| Layer Normalization (layer_normalization_6) | (None, 2, 1024) | 2048 | ['multi_head_attention_6 [0][0]'] |
| Dropout (dropout_6) | (None, 2, 1024) | 0 | ['layer_normalization_6 [0][0]'] |
| LSTM (lstm_3) | (None, 2, 512) | 3147776 | ['dropout_6 [0][0]'] |
| Concatenate (concatenate_3) | (None, 2, 1024) | 0 | ['time_distributed_4[0][0]', 'lstm_3[0][0]'] |
| Multi-head Attention (multi_head_attention_7) | (None, 2, 1024) | 2099712 | ['lstm_3 [0][0]', 'lstm_3 [0][0]'] |
| Layer Normalization (layer_normalization_7) | (None, 2, 1024) | 2048 | ['multi_head_attention_7 [0][0]'] |
| Dropout (dropout_7) | (None, 2, 1024) | 0 | ['layer_normalization_7 [0][0]'] |
| Time Distributed (time_distributed_5) | (None, 2, 1) | 1025 | ['dropout_7 [0][0]'] |

We selected an epsilon value of 1e-6, sufficiently small to minimize its impact on normalization results while ensuring computational stability. Subsequently, after each Layer Normalization Layer, a Dropout Layer is employed to prevent overfitting, which is set to 0.1. The Time Distributed Layer facilitates distributed processing of output at each time step. Two LSTM Layers in the model receive the output from the Time Distributed Layer and the Dropout Layer. Ultimately, the Time Distribution Layer and the Dense Layer collaborate to perform the functions of the Fully Connected Layer. *Relu* serves as the activation function in the encoder, while *Linear* activation is utilized in the decoder. Subsequently, the final result is output in the format (None, 2, 1).

## Performance evaluation

To illustrate the model's regression effect, we selected four pieces of music from different song categories in the Emomusic dataset. As shown in Figs 6 and 7, the model could achieve convergence by training 100 epochs in two emotional dimensions. The regression effect of various music genres was acceptable; the error of the true and predicted values within 1%-3% accounts for more than 90% of the total data.

However, it is worth noting that the less effective regression was usually found at the beginning of each song annotation and in the position where the music-induced emotional ups and downs were more significant. It can be seen that the 15.5s-17s of Song ID = 460, the 15.5s-16.5s, and 40s-43s of Song ID = 214 in the arousal curve, and the 42s-45s of Song ID = 791 in the valence curve were reflected, and the errors of the true value and the predicted value were about 10%.

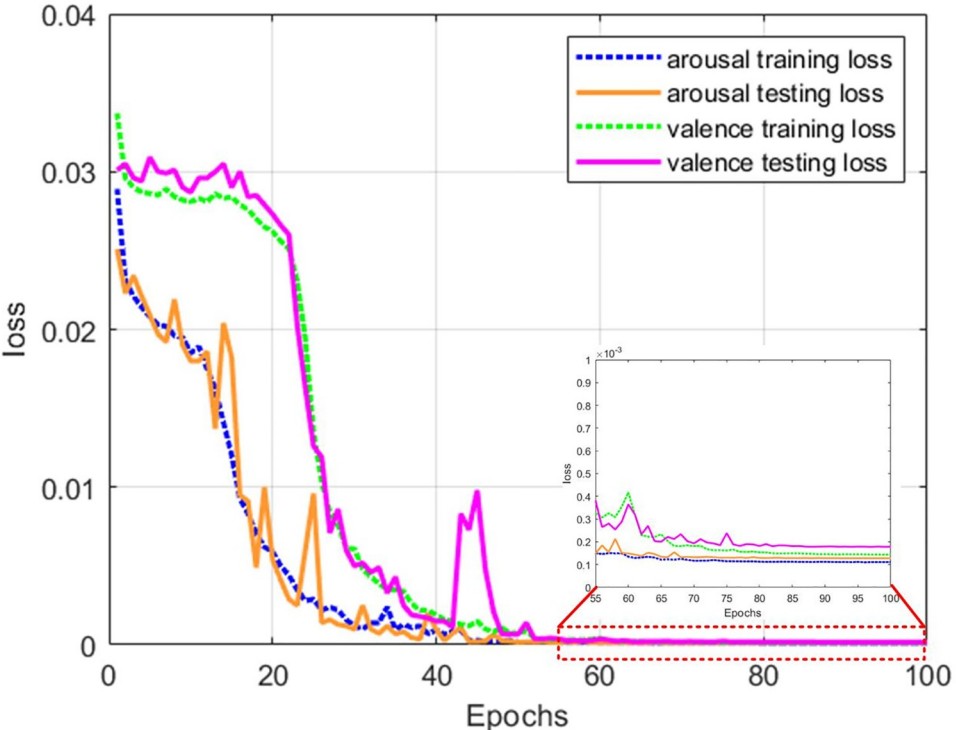

**Fig 6. The relationship curve of emotional dimensions and the information loss rate.**

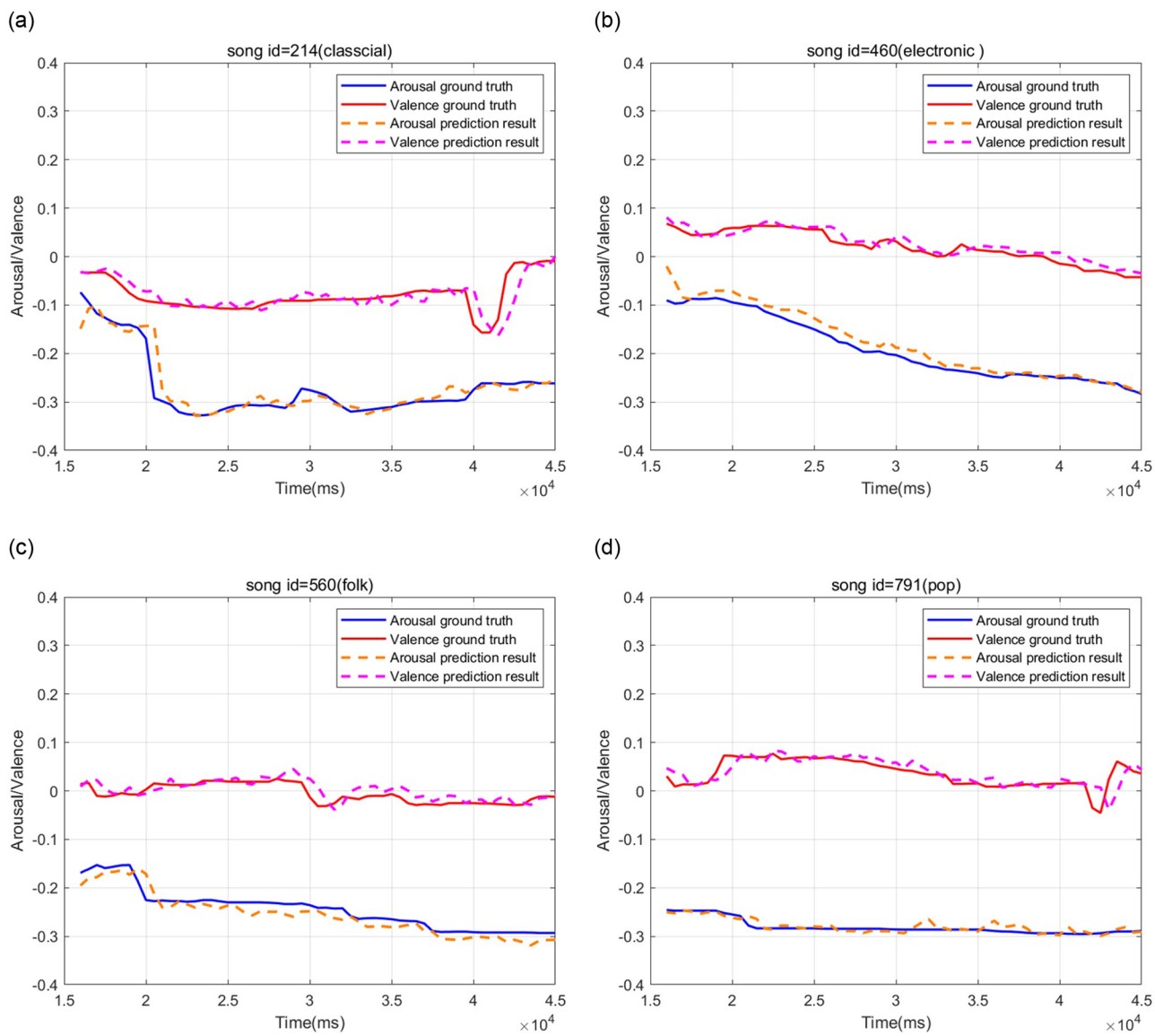

**Fig 7. Regression effect for four random pieces of music in Emomusic dataset(*window size = 2, batch size = 128*).** (a) pop music (song id = 214), (b) electronic music (song id = 460), (c) folk music (song id = 560), (d) pop music (song id = 791).

In addition, we found that although there was a significant error in the position of music-induced emotional fluctuations, in terms of natural curves, the prediction curves generally had a similar change trend and had a delay effect in the time domain. Two reasons for the considerable error were taken into account: On the one hand, the input of the model itself was to splice all the music into a time series, and different music generally had different initial emotional values between them, which led to a low correlation between different sequences. On the other hand, the change of mood required the accumulation of the action time of the musical elements, which made the current musical elements unable to convey the current emotional value accurately. Our explanation was validated in the ablation experiments in the next section.

## Experimental results

### The effect of "*window size*" on regression accuracy

We started with *sequence length* = 2. "*Window size*" represents the size of the window in time series data processing, which determines the size of the time window used to create the training samples. A larger *window size* can capture longer-term time patterns and trends; conversely, a smaller *window size* can better capture short-term patterns and rapid changes. We performed 50 iterative experiments on *window size* = 2, 5, 10, 20, 30, and 60 for each emotion dimension in the Emomusic dataset to evaluate the impact of different *window size* on the regression accuracy.

As shown in Table 3, we found that the regression accuracy of the regression of arousal and valence dimensions was the highest when *window size* = 2 and the regression accuracy was also maintained at a high level when *window size* = 5. Whether it is arousal or valence, the regression accuracy decreased dramatically as the *window size* increased. Comparatively, the arousal dimension exhibits higher prediction accuracy than the valence dimension. However, when the *window size* equals the duration of a single music sequence, both the arousal and valence dimensions demonstrate similar precision.

### The effect of "*batch size*" on regression accuracy

When the model was trained, we found that choosing a larger *batch size* would make the model fall into the local optimal solution, which made the regression accuracy unstable. In order to find the most suitable *batch size*, we investigated the regression precision of *batch size* = 16, 32, 64, 128, 256, 512 in the test set in the Emomusic dataset (Baseline: *window size* = 2), all other things being equal. As with the study of "the effect of *window size*", RMSE and MAE of the model were evaluated in each emotional dimension.

In Table 4, the two emotional dimensions had the most petite MAE and RMSE at *batch size* = 128, and *batch size* < 128 had a worse regression precision than *batch size* > 128. Compared with arousal, valence had better regression stability.

**Table 3. The effect of "*Window size*" on regression accuracy (*sequence length* = 2).**

| Metrics | Size | 2 | 5 | 10 | 20 | 30 | 60 |
|---|---|---|---|---|---|---|---|
| Arousal | RMSE | .0191±.0012 | .0212±.0014 | .0235±.0022 | .0348±.0032 | .0549±.0055 | .3287±.0064 |
|  | MAE | .0141±.0014 | .0157±.0011 | .0169±.0028 | .0246±.0026 | .0442±.0041 | .2800±.0063 |
| Valence | RMSE | .0219±.0014 | .0235±.0018 | .0282±.0027 | .0503±.0042 | .0815±.0045 | .3286±.0029 |
|  | MAE | .0161±.0013 | .0171±.0017 | .0192±.0025 | .0387±.0046 | .0580±.0052 | .2802±.0033 |

**Table 4. The effect of "*Batch size*" on regression accuracy (*sequence length* = 2).**

| Metrics | Size | 16 | 32 | 64 | 128 | 256 | 512 |
|---|---|---|---|---|---|---|---|
| Arousal | RMSE | .2258±.0107 | .2153±.0087 | .0713±.0049 | .0191±.0012 | .0252±.0049 | .0383±.0073 |
|  | MAE | .1817±.0098 | .1726±.0071 | .0571±.0058 | .0141±.0014 | .0180±.0054 | .0295±.0054 |
| Valence | RMSE | .2345±.0050 | .2316±.0013 | .2230±.0010 | .0219±.0014 | .0286±.0038 | .0415±.0061 |
|  | MAE | .1937±.0074 | .1893±.0016 | .1807±.0008 | .0161±.0013 | .0214±.0046 | .0329±.0051 |

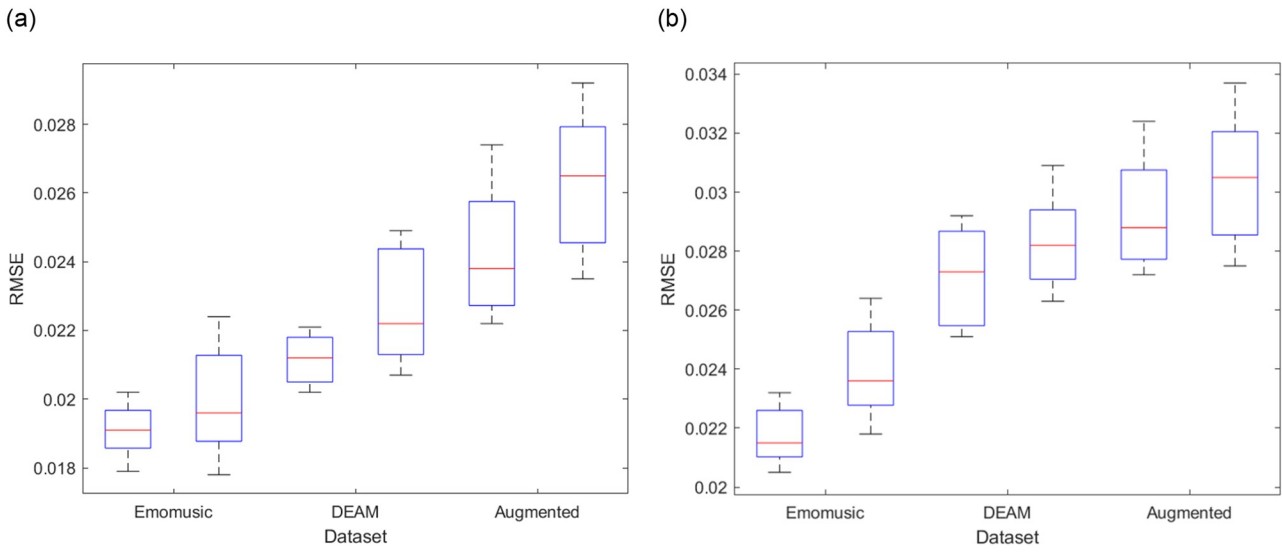

**Fig 8. Comparison of precision box diagrams with and without the Mel-spectrogram features for different datasets (a represents arousal dimension, b represents valence dimension.** In each dataset, the original model is on the left, and the model without the Mel-spectrogram features is on the right).

## The necessity of the Mel-spectrogram features to be extracted from the Efficient Net

In order to prove that the Mel-spectrogram features extracted by Efficient Net are necessary in the second-level regression task, we removed these features from the model in the regression task. While other conditions remained constant, the model was run 50 iterations in each emotion dimension of each dataset.

As shown in Fig 8, in general, the regression effect of the arousal dimension was better than the valence dimension. However, the training effect of the arousal dimension was not as stable as the valence dimension.

Among the three datasets of the original model, the Emomusic dataset had the best regression effect, with RMSE of 0.0191±0.0012, 0.0219±0.0014 for the arousal dimension and valence dimension, respectively; 0.0212±0.0010, 0.0272±0.0020 for the DEAM dataset; and 0.0248 ±0.0026, 0.0298±0.0026 for the Augmented dataset. In addition, we found that the removal of the Mel-spectrogram features significantly reduced the training accuracy. It was evident in the Augmented dataset, where the RMSE of the arousal and valence dimensions increased by at least 0.0024, 0.0019, which fully illustrated the necessity of the Mel-spectrogram features in the regression task.

## The ability of the ENMI model to predict long sequences

The above research was limited to the prediction of the length of the sequence in seconds. However, in practical applications, we may need to predict the emotion flow induced by music at one time; this requires our model to have the ability to predict long sequences. As shown in Figs 9 and 10, the ability of the model to predict different sequence lengths when *window size* = 2 is shown (when testing *sequence length* = 60, to ensure that the *window size* = 2 remains unchanged, *sequence length* = 59 is used instead). The measured statistics were RMSE and Epoch numbers.

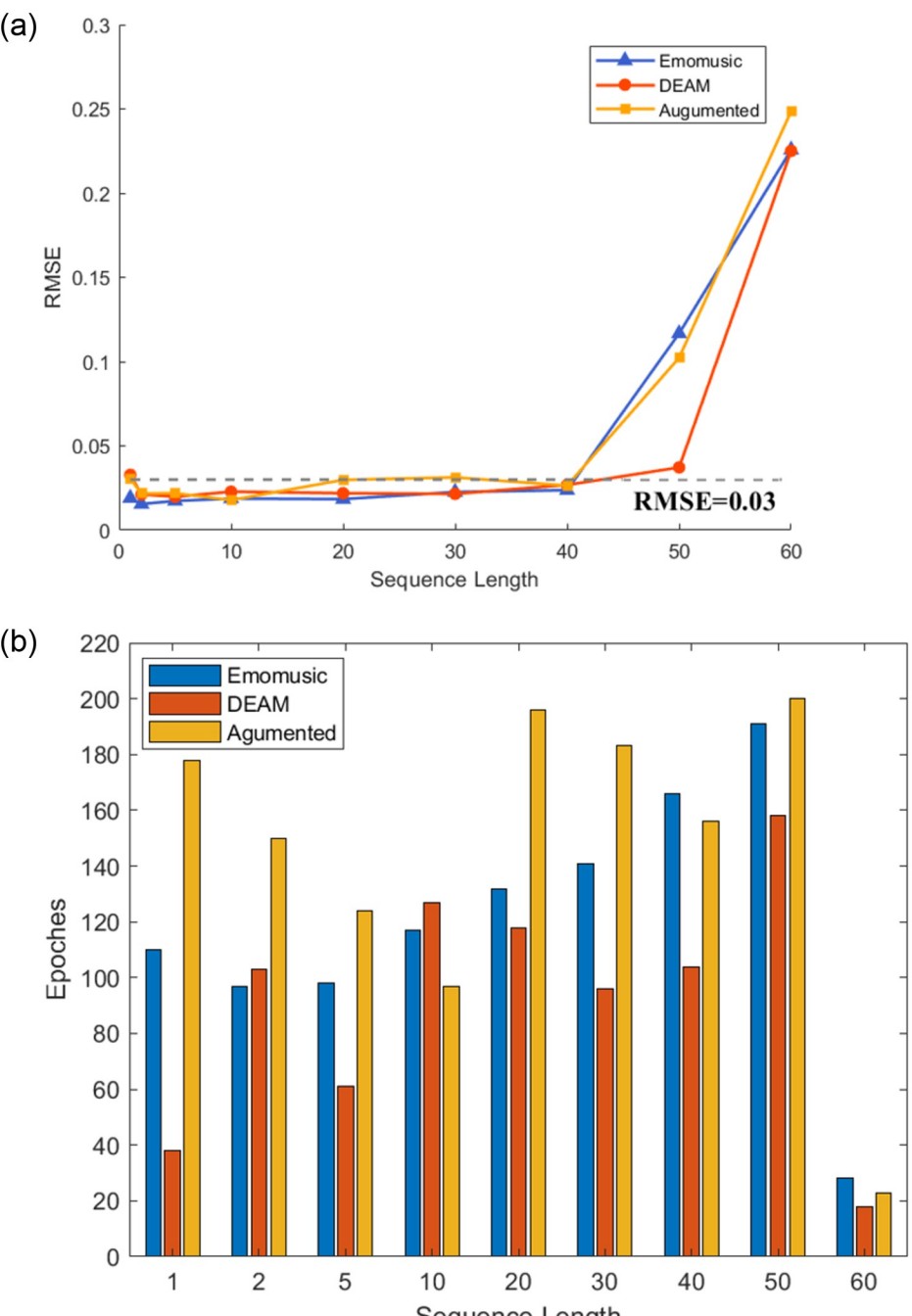

**Fig 9. Performance of the arousal dimension under different prediction sequence lengths in three datasets (a represents the accuracy for different sequence lengths, b represents the epoch numbers for different sequence lengths).**

Contrary to our belief, the regression precision of *sequence length* = 1 was not optimal. As shown in Fig 9a, the performance of *sequence length* = 1 in the three datasets was not as good as *sequence length* = 2, and the RMSE of the DEAM dataset and the Augmented dataset was higher than 0.03; the performance of the Augmented dataset in *sequence length* = 1 was even

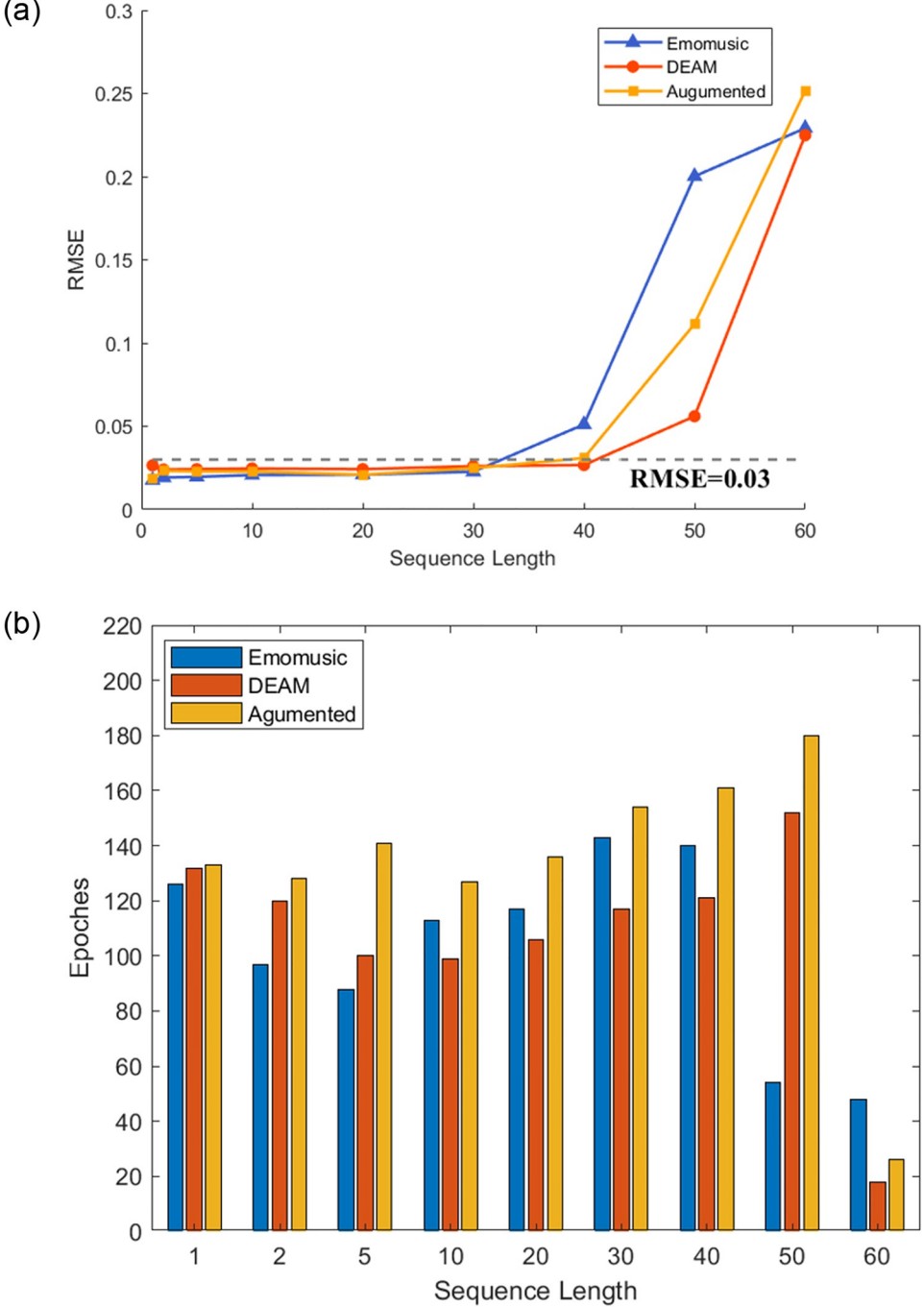

**Fig 10. Performance of the arousal dimension under different prediction sequence lengths in three datasets (a represents the accuracy for different sequence lengths, b represents the epoch numbers for different sequence lengths).**

equivalent to *sequence length* = 20 and 30. This showed that the emotion flow could undergo abrupt changes under the influence of specific musical elements, and the model was less capable of capturing this abrupt change, which was fully confirmed by the regression effect in Fig 7. On the contrary, the model had excellent medium-long distance prediction capabilities. As

shown in Fig 9a and 9b, in the two emotional dimensions, the RMSE had no apparent upward trend in the interval of *sequence length* = [2, 40] and generally remained below 0.03.

In addition, we also found that data augmentation can effectively improve the regression accuracy of the medium-long range of the Emomusic dataset in the valence dimension. However, it played the opposite role in the arousal dimension. Combined with Figs 9a and 10b, in addition to *sequence length* = 60, the epoch numbers of the three datasets tended to increase with the increase of sequence length. By comparing the three datasets, the Augmented dataset required more epochs. The DEAM dataset required fewer epochs and could achieve the best regression accuracy.

## The effect of music feature on regression accuracy

In deep learning, irrelevant and redundant features can lead to inaccurate conclusions. We used the Music Informer model to discuss which features dominate the regression task in each dataset. Similar work has been done by Yang et al. [9] and Dong et al. [26]. They evaluated the model using different feature sets. Zhou et al. [47] evaluated the ability of each hand-extracted musical feature to classify discrete emotion labels and concluded that the MFCC was considered the most effective of all musical features.

Different from their conclusions, as shown in Tables 5 and 6 (3–4 top-ranked values, either MSE or MAE, are highlighted in bold), MFCC features alone participated in the music emotion regression task, and their performance was not outstanding in any emotional dimension in any dataset. The regression analysis employing music theory features demonstrates superior performance compared to MFCC features across two dimensions in each dataset. Music theory features emerge as the most appropriate for regression tasks, particularly within the DEAM dataset, alongside zero Crossings. Among the three music theory features, tempogram and tonal centroid exhibit prominence in the DEAM dataset, albeit showing slightly diminished effectiveness in the Emomusic dataset.

Notably, certain music features (spectral centroid, roughness, spectral bandwidth) exhibit limited capability in predicting continuous sentiment within the Emomusic dataset, potentially attributed to dataset sample size and label quality. Additionally, zero crossings exhibit consistent performance across all dimensions and datasets. While spectral flatness alone demonstrates instability in predicting the arousal dimension, it performs well in the valence dimension.

**Table 5. The regression accuracy of the arousal dimension using individual musical features in the Music Informer model in three datasets(***window size* = 10, *sequence length* = 10).

| Dataset / Feature | Emomusic | | DEAM | |
|---|---|---|---|---|
| | RMSE | MAE | RMSE | MAE |
| MFCC | .0375±.0108 | .0288±.0098 | .0335±.0017 | .0236±.0012 |
| Spectral Centroid | .2948±.0004 | .2486±.0002 | .0506±.0204 | .0359±.0171 |
| Spectral Flatness | .0716±.0492 | .0532±.0387 | .0309±.0031 | .0209±.0030 |
| Zero Crossings | **.0229±.0004** | **.0153±.0005** | **.0284±.0004** | **.0186±.0004** |
| Roughness | .2951±.0002 | .2486±.0001 | .0343±.0028 | .0241±.0035 |
| Spectral Contrast | .0338±.0039 | **.0257±.0044** | .0333±.0018 | .0235±.0020 |
| Spectral Bandwidth | .2944±.0008 | .2479±.0007 | .0306±.0012 | .0205±.0013 |
| Chromaticity Characteristics | **.0246±.0006** | **.0174±.0004** | .0296±.0020 | .0202±.0026 |
| Tempogram | **.0325±.0061** | .0258±.0068 | **.0281±.0005** | **.0181±.0005** |
| Tonal Centroid | .0755±.0476 | .0581±.0363 | **.0281±.0004** | **.0181±.0004** |

**Table 6. The regression accuracy of the valence dimension using individual musical features in the Music Informer model in three datasets(*window size* = 10, *sequence length* = 10).**

| Dataset<br>Feature | Emomusic | | DEAM | |
|---|---|---|---|---|
| | RMSE | MAE | RMSE | MAE |
| MFCC | .0340±.0056 | .0261±.0055 | .0351±.0008 | .0247±.0006 |
| Spectral Centroid | .2935±.0020 | .2476±.0015 | .1590±.1257 | .0237±.0004 |
| Spectral Flatness | **.0252±.0002** | **.0166±.0003** | **.0310±.0003** | **.0197±.0004** |
| Zero Crossings | **.0253±.0003** | **.0166±.0003** | **.0312±.0004** | **.0202±.0004** |
| Roughness | .2949±.0002 | .2487±.0001 | .0355±.0034 | .0235±.0039 |
| Spectral Contrast | .0358±.0035 | .0257±.0024 | .0359±.0023 | .0252±.0020 |
| Spectral Bandwidth | .1626±.1331 | .1345±.1050 | .0329±.0016 | .0222±.0018 |
| Chromaticity Characteristics | **.0263±.0004** | **.0180±.0005** | .0325±.0018 | .0220±.0021 |
| Tempogram | .0276±.0011 | .0194±.0011 | **.0319±.0004** | **.0212±.0005** |
| Tonal Centroid | .0309±.0052 | .0223±.0052 | **.0311±.0005** | **.0201±.0005** |

## Contrast experiment

To further validate the effectiveness of the ENMI model, we compared it with other leading dynamic music emotion recognition models using the same evaluation metrics. As shown in Tables 7 and 8, the ENMI model exhibits the lowest RMSE across any two dimensions of the two datasets, indicating its superior regression performance. Below is a description of the models considered in the comparison for dynamic music emotion recognition using regression methods:

**Table 7. RMSE of dynamic music emotion results of different models on the Emomusic dataset.**

| Model | Arousal(RMSE) | Valence(RMSE) |
|---|---|---|
| CBSA [29] | 0.0725 | 0.0825 |
| RND [48] | 0.2500±0.1300 | 0.2300±0.1100 |
| BSL [48] | 0.2500±0.1100 | 0.2300±0.1000 |
| TUM [48] | 0.0800±0.0500 | 0.0800±0.0400 |
| MLR [49] | 0.1200 | 0.1500 |
| SVR [49] | 0.1000 | 0.1200 |
| GPR [49] | 0.1000 | 0.1200 |
| ConvNet_D-SVM [50] | 0.1000 | 0.0825 |
| CNN-BiLSTM [51] | 0.0700±0.0500 | 0.0600±0.0400 |
| ENMI (**Our model**) | **0.0221±0.0044** | **0.0246±0.0041** |

**Table 8. RMSE of dynamic music emotion results of different models on the DEAM dataset.**

| Model | Arousal(RMSE) | Valence(RMSE) |
|---|---|---|
| BLSTM-RNN [22] | 0.2320±0.1040 | 0.3140±0.0990 |
| Deep LSTM-RNN [22] | 0.2140±0.0570 | 0.2530±0.2010 |
| BCRSN [26] | 0.1010±0.0160 | 0.1230±0.1010 |
| CBSA [29] | 0.0795 | 0.0784 |
| ResNets-audioLIME [52] | 0.2500 | 0.2100 |
| AC2DConvstat [53] | 0.2003 | 0.1928 |
| ENMI (**Our model**) | **0.0440±0.0238** | **0.0352±0.0101** |

BLSTM-RNN, Deep LSTM-RNN [22]: These models are recognized as the two leading methods in MediaEval 2015. Baseline features are derived from the DEAM dataset, consisting of 260 low-level features extracted using the Opensmile Tool. Through PCA whitening (retaining 99% variance), the original audio signals, initially with 2646 dimensions per frame (60ms), are reduced to 43. Subsequently, after dimensionality reduction, audio signals within each 0.5s segment are sequentially fed into these models frame by frame.

RND, BSL, TUM [48]: The Technische Universität München (TUM) team's approach is based on supra-segmental features derived from statistical functions applied to framewise low-level descriptors (LLDs) across either one-second segments or entire songs. (Abbreviations: RND for random level, BSL for Baseline)

MLR, SVR, GPR [49]: These three models serve as the dataset baseline, while Aizu University and Utrecht University employ their respective recognition methods for training and evaluating the EmoMusic dataset.

ConvNet_D-SVM [50]: This study explores the integration of contextual information into emotional analysis by utilizing dilated convolutions to expand the receptive fields of network layers and then feeding this information into an SVM regression model.

CNN-BiLSTM [51]: Dynamic music emotion recognition is achieved by integrating CNNs and BiLSTM Networks.

ResNets-audioLIME [52]: The source separation interpreter and the residual network are integrated to analyze both intermediate perceptual features and spectrogram features.

AC2DConvstat [53]: Employing a two-dimensional CNN model, both in audio and computational domains, to analyze the audio feature representations formed from a fusion of raw audio, audio signals, and spectrograms.

## Conclusion and future work

This paper transforms the problem of continuous emotion recognition in music into a multivariate time series regression problem. We propose a lightweight time series prediction model, the ENMI model, to effectively explore the continuous mapping relationship between music and emotional flow. Hence, a hierarchical classification approach was used to manually extract time series features from the different dimensions of the audio file. The time series features are distinctive and may lead to overfitting. Therefore, they were augmented using the Efficient Net that extracts music features from the spectrogram of the Mel of the audio file. Ablation experiments demonstrated that this augmentation further enhanced regression accuracy. Additionally, various hyperparameters of the Music Informer model were examined. Consequently, an optimal solution was identified. Finally, to validate the effectiveness of the model, prediction tests with different sequence lengths were conducted, while the most effective features were discussed.

Evaluation of the regression performance of the developed model revealed that the prediction accuracy for long-term emotion flow was significantly lower compared to second-level predictions. This may be due to the manner through which time data is aligned such that the correlation between the samples of music before and after alignment is significantly low. Future research will focus on solving the "timestamp" issues related to music emotion sequences and explore the relationship between music and emotions on a larger scale. Additionally, while MER focuses on how computers perceive the emotional impact of music

samples, it fails to account for individual differences. Future studies should be geared towards building a larger dataset incorporating a wider range of music samples and test subjects to analyze the general trends. It is anticipated that this effort will be quite significant, especially for advancing practical applications in Music Information Retrieval (MIR) and approaches to music therapy.

## Supporting information

**S1 File. Minimal dataset.**
(ZIP)

**S2 File. The points extracted from images for analysis.**
(ZIP)

**S3 File. The values behind the means, standard deviations and other measures reported.**
(ZIP)

**S4 File. The values used to build graphs.**
(ZIP)

**S5 File. All prediction results.**
(ZIP)

## Acknowledgments

We would like to thank Mohammad Soleymani et al. for providing the Emomusic dataset and Anna Aljanaki et al. for providing the DEAM dataset. Without them, this study would not have been possible.

## Author Contributions

**Conceptualization:** Yunrui Shang.

**Data curation:** Yunrui Shang.

**Formal analysis:** Yunrui Shang.

**Investigation:** Qi Peng.

**Methodology:** Yunrui Shang.

**Project administration:** Yunrui Shang.

**Software:** Zixuan Wu.

**Supervision:** Yinhua Liu.

**Validation:** Yunrui Shang.

**Visualization:** Yunrui Shang.

**Writing – original draft:** Yunrui Shang.

**Writing – review & editing:** Yunrui Shang.

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
