## [Decision Letter · Decision Letter 0]

27 Feb 2024

PONE-D-24-01232Music-induced emotion flow modeling by ENMI NetworkPLOS ONE

Dear Dr. Liu,

Thank you for submitting your manuscript to PLOS ONE. After careful consideration, we feel that it has merit but does not fully meet PLOS ONE’s publication criteria as it currently stands. Therefore, we invite you to submit a revised version of the manuscript that addresses the points raised during the review process.

Please ensure that you respond to all review requirements. Focus specifically on the following aspects:

Improve and update the literature review sectionIn your response, please address the challenges and shortcomings of the methods used in existing studies. Additionally, explain the reasons for selecting the chosen model and highlight the advantages it provides.Additional implementation details for the selected methods, including a description of the database used and the hyperparameters applied, should be provided.Specify the limitations of this studyIndicate of future study directionsPlease ensure that your writing is clear and free of any spelling, grammar, or punctuation errors.Pay attention to the correct use of certain notions, also taking into account the indications of the reviewersUnification of the format of bibliographic references ==============================

We look forward to receiving your revised manuscript.

Kind regards,

Livia Petrescu

Academic Editor

PLOS ONE

Journal Requirements:

2. Thank you for submitting the above manuscript to PLOS ONE. During our internal evaluation of the manuscript, we found significant text overlap between your submission and previous work in the [introduction, conclusion, etc.].

Please revise the manuscript to rephrase the duplicated text, cite your sources, and provide details as to how the current manuscript advances on previous work. Please note that further consideration is dependent on the submission of a manuscript that addresses these concerns about the overlap in text with published work.

[If the overlap is with the authors’ own works: Moreover, upon submission, authors must confirm that the manuscript, or any related manuscript, is not currently under consideration or accepted elsewhere. If related work has been submitted to PLOS ONE or elsewhere, authors must include a copy with the submitted article. Reviewers will be asked to comment on the overlap between related submissions (http://journals.plos.org/plosone/s/submission-guidelines#loc-related-manuscripts).]

We will carefully review your manuscript upon resubmission and further consideration of the manuscript is dependent on the text overlap being addressed in full. Please ensure that your revision is thorough as failure to address the concerns to our satisfaction may result in your submission not being considered further.

"The research was funded by the National Key Research and Development Project

(Grant Number: 2020YFB1313604).

We would like to thank Mohammad Soleymani et al. for providing the Emomusic

dataset and Anna Aljanaki et al. for providing the DEAM dataset. Without them, this

study would not have been possible"

5. We note that your Data Availability Statement is currently as follows: [All relevant data are within the manuscript and its Supporting Information files.]

Reviewers' comments:

Reviewer's Responses to Questions

**Comments to the Author**

1. Is the manuscript technically sound, and do the data support the conclusions?

Reviewer #1: Yes

Reviewer #2: Yes

2. Has the statistical analysis been performed appropriately and rigorously? 

Reviewer #1: Yes

Reviewer #2: Yes

3. Have the authors made all data underlying the findings in their manuscript fully available?

Reviewer #1: No

Reviewer #2: Yes

4. Is the manuscript presented in an intelligible fashion and written in standard English?

Reviewer #1: Yes

Reviewer #2: Yes

5. Review Comments to the Author

Reviewer #1: Although the manuscript tries to address an interesting issue, there are some major points that the authors must address to make it suitable for publication.

1- The manuscript lacks literature review section. The authors must discuss existing studies in this field in a separate section and clearly mention the challenges of previous methods.

2- Section 2 is hard to follow and understand. Authors have to provide more information and make the equations more comprehensible for authors.

3- Section 3 must be unified. One sentence cannot be considered as a separate paragraph.

4- Authors are suggested to provide more information about the implementation process and used hyper-parameters and datasets.

5- There are couple of grammatical and verbal mistakes; the authors have to carefully reread the manuscript.

6- Please discuss the motivation behind the proposed model clearly. Mention what is the priority of the proposed model compared to previous studies

7- The authors are suggested to provide some comments on future directions.

8- The format of references is not compatible and the same. Try to carefully update the format of references and use more update references.

9- The following articles are recommended for use in your manuscript:

• Zeinab Khodaverdian, et.al. (2023) An energy aware resource allocation based on combination of CNN and GRU for virtual machine selection. Multimedia Tools and Applications Multimedia Tools and Applications, 2023.

https://doi.org/10.1007/s11042-023-16488-2

• Sadr H, Nazari Soleimandarabi M (2022) ACNN-TL: attention-based convolutional neural network coupling with transfer learning and contextualized word representation for enhancing the performance of sentiment classification. J Supercomputer 78(7):10149–10175.

https://doi.org/10.1007/s11227-021-04208-2

• Fatemeh Mohades Deilami, et.al. (2022) Contextualized Multidimensional Personality Recognition using Combination of Deep Neural Network and Ensemble Learning. Neural Process Letters, 54(5), pp. 3811–3828.

https://doi.org/10.1007/s11063-022-10787-9

• Kalashami, M.P., et.al. (2022) EEG Feature Extraction and Data Augmentation in Emotion Recognition. Computational Intelligence and Neuroscience, 2022, 7028517.

https://doi.org/10.1155/2022/7028517

• Hossein Sadr, et.al. (2019) Exploring the Efficiency of Topic-Based Models in Computing Semantic Relatedness of Geographic Terms, International journal of web research, Volume 2, Issue 2, Autumn-Winter, December 2019, Pages 23-35.

https://doi.org/10.22133/ijwr.2020.225866.1056

Reviewer #2: The authors proposed a novel architecture for MER by parallely processing the audio features in two independent time scales. The proposed method is evaluated on two standard benchmark datasets. Althought the proposed solution is interesting, I see some flaws they need to be resolved. My comments are below:

- The proposed solution is thoroughly evaluated on two publicly available benchmark datasets, however a comparison with other studies and systems is missing (several benchmark tests and studies on the same datasets have been published recently, e.g. MediaEval's Emotion in Music challenge, [19] /in the references, and later ones).

- The related work review in the Introduction should be updated - the latest cited work on music emotion prediction in continuous scale is [19], which is from 2019 – more recent works (if there are some) needs to be added.

- Please also clarify the statement (p.2): " Although good progress has been made, *the models are unstable*" I do not agree that the classical probability based machine learning models (such as HMM or SVM) are unstable for MER tasks.

Mentioning a *RReliefF* method in the related work review (p2) seems be redundant.

- This question arises from the first comment: Did you apply the same training/testing methodology as is recommended for the DEAM dataset? [15,19]. This is necessary in order to make a comparison with the other solutions proposed recently (if they make evaluation on the same data).

- Please clarify, why you decided to use mean absolute error (MAE) as an additional metric to RMSE. Cross-Correlation is more common for time series comparison (cross-correlation based metrics were also use in the Mediaeval Emotion in Music benchmark [15].

- p.11, fig 5 - I do not understand why mp3 files are converted to flac format, thus, from lossy to lossless one. If there is already lossy compression, transformation to lossless format has no benefit. In addition, flac format is not suitable for data manipulation. If the reason was to have a format for easy access of data, raw or wav format suits better than flac. I recommend, withdraw information about flac from the text. In figure, replace the label *.flac with "raw audio".

- p. 18-19, Tables 4,5: I recommend to highlight (e.g. by bold font) the best results for all columns (e.g. 4 top ranked values either MSE or MAE) to better visualize the most salient features.

- p.2 - "*invisible* Markov model" - correct is "hidden Markov model"

- p.12 - " .. spectral correlation features (including MFCC ... " - the word 'correlation' is confusing. Instead, use simply

"spectral features".

- Regarding the name "non-time series": I do not think that Mel spectrogram features are non-time series, beacuse one dimension of the spectrogram is time, just there is a different time-scale as in the case of selected features extracted by Librosa lib. I recommend to use a name for the Mel-spectrogram features other than "non-time".

- p.13 What difference is between "sycles"and "epochs"? Please, unify this.

- p. 14 - Fig 6 capture does not match with graphs in Figure 6.

- p.6, 7 - Please cite references for both the Effcient Net model and Music Informer.

- Please provide a source, from which the DEAM and Emomusic datasets can be accessed.

6. PLOS authors have the option to publish the peer review history of their article (what does this mean?). If published, this will include your full peer review and any attached files.

Reviewer #1: No

Reviewer #2: No

---

## [Author Response · Author response to Decision Letter 0]

25 Apr 2024

Dear Editors and Reviewers:

 Thank you very much for giving us an opportunity to revise our manuscript. We appreciate the editors and reviewers very much for your constructive comments and suggestions on our manuscript entitled “Music-induced emotion flow modeling by ENMI Network”

 Upon receiving your notification, we promptly delved into the reviewer's comments and suggestions, recognizing the depth of their insights into both the specific sections and the overall structure of the paper. The constructive feedback not only offers a fresh perspective but also provides valuable directions for refining the manuscript. Understanding the stringent academic standards set by PLOS ONE, we are dedicated to revising the paper to meet the highest standards and ensure it attains optimal academic excellence.

 I sincerely appreciate your and the reviewer's recognition of my work, and I look forward to aligning the revised manuscript more closely with the expectations of the journal. I will work diligently to address the reviewer's comments comprehensively, aiming to make a positive contribution towards the paper's ultimate acceptance.

 Best regards,

 Lead author: Yunrui Shang 

 E-mail: m15031528057@163.com

 Corresponding author: Yinhua Liu

 E-mail: liuyinhua@qdu.edu.cn

---

## [Decision Letter · Decision Letter 1]

4 Jul 2024

PONE-D-24-01232R1Music-induced emotion flow modeling by ENMI NetworkPLOS ONE

Dear Dr. Liu,

Thank you for submitting your manuscript to PLOS ONE. After careful consideration, we feel that it has merit but does not fully meet PLOS ONE’s publication criteria as it currently stands. Therefore, we invite you to submit a revised version of the manuscript that addresses the points raised during the review process.

Kindly respond comprehensively to each of Reviewer 2's comments and specify any modifications made to the manuscript in response to these critiques. Please indicate the exact locations of these changes and provide a brief description of the improvements implemented.

In addition, please clarify the requirements of Reviewer 3, including:

Preprocessing of data,Definition and explanation of EquationsThe grouping of methodological descriptions (many of the reports from the discussions are actually results) Please submit your revised manuscript by Aug 18 2024 11:59PM. If you will need more time than this to complete your revisions, please reply to this message or contact the journal office at plosone@plos.org. Please include the following items when submitting your revised manuscript:A rebuttal letter that responds to each point raised by the academic editor and reviewer(s). You should upload this letter as a separate file labeled 'Response to Reviewers'.A marked-up copy of your manuscript that highlights changes made to the original version. You should upload this as a separate file labeled 'Revised Manuscript with Track Changes'.An unmarked version of your revised paper without tracked changes. You should upload this as a separate file labeled 'Manuscript'.

We look forward to receiving your revised manuscript.

Kind regards,

Livia Petrescu

Academic Editor

PLOS ONE

Reviewers' comments:

Reviewer's Responses to Questions

**Comments to the Author**

1. If the authors have adequately addressed your comments raised in a previous round of review and you feel that this manuscript is now acceptable for publication, you may indicate that here to bypass the “Comments to the Author” section, enter your conflict of interest statement in the “Confidential to Editor” section, and submit your "Accept" recommendation.

Reviewer #1: All comments have been addressed

Reviewer #2: (No Response)

Reviewer #3: (No Response)

2. Is the manuscript technically sound, and do the data support the conclusions?

Reviewer #1: Yes

Reviewer #2: Partly

Reviewer #3: Partly

3. Has the statistical analysis been performed appropriately and rigorously? 

Reviewer #1: Yes

Reviewer #2: N/A

Reviewer #3: Yes

4. Have the authors made all data underlying the findings in their manuscript fully available?

Reviewer #1: Yes

Reviewer #2: No

Reviewer #3: Yes

5. Is the manuscript presented in an intelligible fashion and written in standard English?

Reviewer #1: Yes

Reviewer #2: No

Reviewer #3: Yes

6. Review Comments to the Author

Reviewer #1: The authors of the article have made a successful attempt to respond to the mentioned issues and therefore the manuscript is accepted for publication.

Reviewer #2: The "Author's Response To Reviewer Comments" letter is too brief and insuffiecient. The authors should responds to each point raised by the academic editor and reviewer(s) in this letter.

Reviewer #3: The paper proposes a method for music emotion recognition that consists in estimating both arousal and valence in a regression framework from features extracted either from the audio signals or their mel spectrograms. The authors employ an Efficient Net to learn features from the mel spectrograms and send them along the manually extracted features to an Informer model. The performances of the proposed approach are evaluated on publicly available datasets (DEAM, Emomusic) and compared to other methods from the related literature. Ablation studies are carried out to verify the importance of all components of the proposed approach.

On the one hand, the methodological aspects of this paper look sound, and the carried out experiments are relatively comprehensive. On the other hand, I think many methodological aspects are currently described in a fuzzy/suboptimal way in the manuscript, and would require some clarification in order for this paper to be ready for publication. More specifically, I would have the following questions and comments:

1- I found the wording of the 1st contribution of the paper (top of page 5) to be somewhat fuzzy. It would be advised to replace some vague wording like "creatively explored" with more specific sentences.

2- Section "Emotion flow on Arousal-Valence emotional space": it is not clear to me what is the "music element vector" $\\delta_y$ in Equation (1). The Pearson correlation being computed between $\\delta_x$ and $\\delta_y$ implies that both vectors have the same dimension, and $\\delta_x$ is assumed to be 2D (vector in the arousal/valence space). It would be required to provide a more detailed definition of $\\delta_y$ to make the understanding of Equation (1) easier.

3- Subsection "Emotion flow on Arousal-Valence emotional space": similarly to the previous point, it is not so clear to me what $f$/$f_A$ and $g$/$g_A$ represent in Equation (2). $f$ and $g$ are introduced as the "laws of the arousal dimension and the valence dimension", which is quite fuzzy to me. $f_A$ and $g_A$ are not introduced at all. It is better to be more specific regarding these definitions as Equation (2) is currently difficult to understand.

4- What does "Ascension" in Figure 3 mean? Since it is currently not described in the manuscript, it would be better to add an explanation there.

5- Subsection "Informer layer": the authors justify their decision to add the song ID in the Time Feature Embedding with the sentence "Considering that the time series between different songs are much less relevant than the same song," which is quite unclear to me. Additionally, this choice raises the question of how the model is used at inference time on unseen data when the song ID is not known (e.g. for a song that doesn't belong to the DEAM or Emomusic datasets, without any ID). A more detailed and clearer explanation of these two points is necessary in my opinion. It could also be interesting to report the results of the proposed approach without using the song ID in the embedding.

6- Subsection "Preprocessing": the first paragraph describing the datasets that were used in the study is not so clear, more specifically regarding to what the "new dataset [introduced] to achieve balance" refers to. The abstract mentions an "augmented version" of the Emomusic dataset, but the wording used in this Subsection seems to imply this augmented version is the DEAM dataset. It would be indicated to clarify this part and how each of the 3 datasets was used (since the experiments were apparently carried out only on Emomusic and DEAM).

7- Subsection "Preprocessing": the sentence "To ensure that the addition of white Gaussian noise ..." seems to be incomplete and should therefore be revised.

8- Subsection "Feature extraction": the description of the features that were manually extracted from the audio time series should be moved to the "Model description" Section in my opinion, as it is part of the methodological description according to Figure 3.

9- The "Discussion" Section could be renamed "Experimental results" as it contains the main results of the study as well as the ablation studies.

7. PLOS authors have the option to publish the peer review history of their article (what does this mean?). If published, this will include your full peer review and any attached files.

Reviewer #1: No

Reviewer #2: No

Reviewer #3: **Yes: **Frédéric Li

---

## [Author Response · Author response to Decision Letter 1]

12 Jul 2024

Dear reviewer #2,

I am deeply honored to have received your review comments, and we are also very grateful for the opportunity to revise our manuscript.

In the last version of the modification our reply was too simple and did not indicate the location of the modification clearly enough, we deeply reflect on this and sincerely apologize to you. In this revision, we have carefully studied your suggestions and replied to each of them, at the same time, in the version of Revised Manuscript with Track Changes, we have shown you the specific location of each of your comments.

Thank you again for your time and for recognizing the manuscript, we have gained a lot from your comments. Please feel free to contact us if you have additional comments on this revision.

Yours sincerely

Lead author: Yunrui Shang 

Corresponding author: Yinhua Liu

Dear Dr.Li(reviewer #3),

It is my honor to receive your review comments and we appreciate the opportunity to revise the manuscript. Your nine comments contain suggestions in three areas: methodology of sentiment modeling, data preprocessing, and some presentation issues with ambiguity.

Your comments make us realize that you are rigorous and serious about your research. We have fully studied your comments and revised the errors and inappropriateness that appeared in the manuscript, and have listed our responses and revisions in detail. Please feel free to contact us with any additional questions regarding the following revisions.

Thank you for reviewing and recognizing our manuscript.

Yours sincerely

Lead author: Yunrui Shang 

Corresponding author: Yinhua Liu

---

## [Decision Letter · Decision Letter 2]

13 Sep 2024

PONE-D-24-01232R2Music-induced emotion flow modeling by ENMI NetworkPLOS ONE

Dear Dr. Liu,

Thank you for submitting your manuscript to PLOS ONE. After careful consideration, we feel that it has merit but does not fully meet PLOS ONE’s publication criteria as it currently stands. Therefore, we invite you to submit a revised version of the manuscript that addresses the points raised during the review process.

Please clarify the requirements of Reviewer 3, including:

**Clarify the First Contribution**: The current wording remains somewhat generic. Please specify that the contribution involves modelling "emotion flow recognition as a regression problem using time series."**Equation (2) Notation**: The use of <math xmlns="http://www.w3.org/1998/Math/MathML"><semantics><mrow><msub><mi>$g_A$  for valence is unclear. Consider changing it to</mi></msub></mrow></semantics></math><math xmlns="http://www.w3.org/1998/Math/MathML"><semantics><mrow><msub><mi>$g_V$ for consistency.</mi></msub></mrow></semantics></math>**Song ID Explanation**: The explanation of "data splicing" remains unclear. Additionally, your approach may only work with specific data formats, which could be a limitation. Please address this in the "Conclusion and Future Work" section..**Dataset Augmentation**: Instead of presenting the augmented Emomusic samples as a separate dataset, clarify that the Emomusic dataset was augmented with white noise and scaling, as described in the "Processing" section.null**Final Check**: Please review the manuscript for any spelling or grammar issues, particularly the redundant sentences before Equation (16). Please submit your revised manuscript by Oct 28 2024 11:59PM. If you will need more time than this to complete your revisions, please reply to this message or contact the journal office at plosone@plos.org. Please include the following items when submitting your revised manuscript:A rebuttal letter that responds to each point raised by the academic editor and reviewer(s). You should upload this letter as a separate file labeled 'Response to Reviewers'.A marked-up copy of your manuscript that highlights changes made to the original version. You should upload this as a separate file labeled 'Revised Manuscript with Track Changes'.An unmarked version of your revised paper without tracked changes. You should upload this as a separate file labeled 'Manuscript'.If applicable, we recommend that you deposit your laboratory protocols in protocols.io to enhance the reproducibility of your results. Protocols.io assigns your protocol its own identifier (DOI) so that it can be cited independently in the future. For instructions see: https://journals.plos.org/plosone/s/submission-guidelines#loc-laboratory-protocols. Additionally, PLOS ONE offers an option for publishing peer-reviewed Lab Protocol articles, which describe protocols hosted on protocols.io. Read more information on sharing protocols at https://plos.org/protocols?utm_medium=editorial-email&utm_source=authorletters&utm_campaign=protocols.

We look forward to receiving your revised manuscript.

Kind regards,

Livia Petrescu

Academic Editor

PLOS ONE

Journal Requirements:

Reviewers' comments:

Reviewer's Responses to Questions

**Comments to the Author**

1. If the authors have adequately addressed your comments raised in a previous round of review and you feel that this manuscript is now acceptable for publication, you may indicate that here to bypass the “Comments to the Author” section, enter your conflict of interest statement in the “Confidential to Editor” section, and submit your "Accept" recommendation.

Reviewer #2: All comments have been addressed

Reviewer #3: (No Response)

2. Is the manuscript technically sound, and do the data support the conclusions?

Reviewer #2: Yes

Reviewer #3: Yes

3. Has the statistical analysis been performed appropriately and rigorously? 

Reviewer #2: Yes

Reviewer #3: N/A

4. Have the authors made all data underlying the findings in their manuscript fully available?

Reviewer #2: Yes

Reviewer #3: Yes

5. Is the manuscript presented in an intelligible fashion and written in standard English?

Reviewer #2: Yes

Reviewer #3: Yes

6. Review Comments to the Author

Reviewer #2: The authors put a lot of effort into thoroughly revising the manuscript. They adequately addressed all the comments.

Reviewer #3: I would first like to thanks the authors for the obvious care they put in addressing my previous comments, and their very thorough response. I think several of my concerns have properly been addressed. I would nevertheless still have questions and remarks on some aspects that I would like to have clarified before I can provide my recommendation to accept the paper.

Comment #1- Wording of the first paper contribution:

The authors have disambiguated the description of the first contribution, but its wording now seems to be somewhat generic to me. I would recommend to more specifically state that the contribution consists in modelling emotion flow recognition as a regression problem using time series.

Comment #3 - Equation (2):

It seems the authors are using A and V as subscript for arousal and valence respectively. The notation $g_A$ to designate the emotion change according to the valence dimension is therefore not so clear. Maybe $g_V$ would be more appropriate.

Comment #5 - song ID:

The authors' answer for this point was not completely clear to me on some aspects. For instance, I am unsure of what "data splicing different songs by the first and last of the time dimension" refers to. Furthermore, from my understanding, a timestamp of the format "ID - second" was used as input of the time embedding layer in the carried out experiments. Because this information may not be available for all songs, the authors seem to indicate in their answer that their proposed approach can be applied only on data following a very specific format. This seems to be a strong limitation to me, that should a minima be discussed in the "Conclusion and future work" section.

Comment #6 - dataset description:

From my understanding of the authors' response, the "Augmented dataset" consists of music samples from the Emomusic dataset, that were corrupted with white noise and scaled to create augmented samples in order to compensate for the lower sample size of Emomusic compared to DEAM. In my opinion, presenting these augmented samples as a third dataset may lead to confusion, because it was jointly used with Emomusic for the carried out experiments. It would be better to mention in the manuscript that the Emomusic dataset was augmented with the techniques described in Section "Processing" to account for its lower size compared to DEAM.

Finally as a minor remark, I would suggest the authors to perform a last spell/grammar check of the modified parts of the manuscript (e.g. redundant sentences in the added paragraph before Equation 16).

7. PLOS authors have the option to publish the peer review history of their article (what does this mean?). If published, this will include your full peer review and any attached files.

Reviewer #2: No

Reviewer #3: **Yes: **Frédéric Li

---

## [Author Response · Author response to Decision Letter 2]

17 Sep 2024

Dear Dr. Li,

Thank you for your approval of our revised manuscript in the last round of review.

After receiving your comments, we made further adjustments to the areas that needed improvement. To make sure there are no careless mistakes left in the paper, we asked a professional editor to polish it. We hope that in the next round of peer review, our manuscript will meet the standards for acceptance and publication.

We appreciate your continued attention and review of our manuscript. We have learned a lot from this process. Please feel free to contact us with any additional questions regarding the following revisions.

Yours sincerely

Lead author: Yunrui Shang 

Corresponding author: Yinhua Liu

---

## [Decision Letter · Decision Letter 3]

30 Sep 2024

Music-induced emotion flow modeling by ENMI Network

PONE-D-24-01232R3

Dear Dr. Liu,

We’re pleased to inform you that your manuscript has been judged scientifically suitable for publication and will be formally accepted for publication once it meets all outstanding technical requirements.

Kind regards,

Livia Petrescu

Academic Editor

PLOS ONE

Additional Editor Comments (optional):

Reviewers' comments:

Reviewer's Responses to Questions

**Comments to the Author**

1. If the authors have adequately addressed your comments raised in a previous round of review and you feel that this manuscript is now acceptable for publication, you may indicate that here to bypass the “Comments to the Author” section, enter your conflict of interest statement in the “Confidential to Editor” section, and submit your "Accept" recommendation.

Reviewer #3: All comments have been addressed

2. Is the manuscript technically sound, and do the data support the conclusions?

Reviewer #3: (No Response)

3. Has the statistical analysis been performed appropriately and rigorously? 

Reviewer #3: (No Response)

4. Have the authors made all data underlying the findings in their manuscript fully available?

Reviewer #3: (No Response)

5. Is the manuscript presented in an intelligible fashion and written in standard English?

Reviewer #3: (No Response)

6. Review Comments to the Author

Reviewer #3: I thank the authors for their detailed answer once again. I consider my comments have all been properly addressed, and therefore recommend the acceptance of the paper.

7. PLOS authors have the option to publish the peer review history of their article (what does this mean?). If published, this will include your full peer review and any attached files.

Reviewer #3: **Yes: **Frédéric Li

---

## [Editor Report · Acceptance letter]

8 Oct 2024

PONE-D-24-01232R3 

PLOS ONE

Dear Dr. Liu, 

I'm pleased to inform you that your manuscript has been deemed suitable for publication in PLOS ONE. Congratulations! Your manuscript is now being handed over to our production team.

Kind regards, 

on behalf of

Dr. Livia Petrescu 

Academic Editor

PLOS ONE